# Memory shapes microbial populations

**Chaitanya S. Gokhale**[1]*, **Stefano Giaimo**[2], **Philippe Remigi**[3]

**1** Research Group for Theoretical Models of Eco-evolutionary Dynamics, Department of Evolutionary Theory, Max-Planck Institute for Evolutionary Biology, Plön, Germany, **2** Department of Evolutionary Theory, Max-Planck Institute for Evolutionary Biology, Plön, Germany, **3** LIPME, Universite de Toulouse, INRAE, CNRS, Castanet-Tolosan, France

* gokhale@evolbio.mpg.de

**Data Availability Statement:** https://github.com/tecoevo/phenotypic_heterogeneity.

**Funding:** This study received funding from the Max-Planck-Gesellschaft received by C.S.G. and S. G. Funding from the French Laboratory of Excellence (Agence Nationale de la Recherche)

## Abstract

Correct decision making is fundamental for all living organisms to thrive under environmental changes. The patterns of environmental variation and the quality of available information define the most favourable strategy among multiple options, from randomly adopting a phenotypic state to sensing and reacting to environmental cues. Cellular memory—the ability to track and condition the time to switch to a different phenotypic state—can help withstand environmental fluctuations. How does memory manifest itself in unicellular organisms? We describe the population-wide consequences of phenotypic memory in microbes through a combination of deterministic modelling and stochastic simulations. Moving beyond binary switching models, our work highlights the need to consider a broader range of switching behaviours when describing microbial adaptive strategies. We show that memory in individual cells generates patterns at the population level coherent with overshoots and non-exponential lag times distributions experimentally observed in phenotypically heterogeneous populations. We emphasise the implications of our work in understanding antibiotic tolerance and, in general, bacterial survival under fluctuating environments.

## Author summary

While being genetically the same, a population of cells can show phenotypic variability even under homogeneous environments. Often advantageous under heterogeneous environments, this *phenotypic heterogeneity* is highly relevant in the studies of antibiotic resistance evolution and cancer resurgence. Numerous theoretical models exist applying a simple model of phenotypic switching. Experimental measurements on phenotypic heterogeneity have increased in precision over the past decade, and the simple models are inadequate to explain the new observations. In this paper, we explore the role of cellular memory as a crucial component of phenotypic switching. We see that memory helps account for the hitherto unexplained observations and fundamentally extend our understanding of phenotypic heterogeneity.

project TULIP (ANR-10-LABX-41) was received by P.R. The funders had no role in study design, data collection and analysis, decision to publish, or preparation of the manuscript.

**Competing interests:** The authors have declared that no competing interests exist.

## Introduction

In an ideal world, living organisms would be able to adapt instantly and reliably to changing environmental conditions to maximise their instantaneous performance. However, conditions may change abruptly and unpredictably, making it ineffective to mount a specific rapid response. Also, some responses require the synthesis of complex molecules (secretion systems or capsules in bacteria) or entering a physiological state (dormancy) that cannot be reverted instantaneously, if need be. Switching to a new phenotype may thus commit the cell to a response that lasts for a specific timescale different from the duration of the changing environment. Besides phenotypic switching, diversity in response rates can result in intricate patterns of phenotypic heterogeneity. We postulate that instantiating memory, i.e. some form of mechanistic regulation of the time spent in a state before switching to another, will affect population composition, size and ultimately fitness. While classical models of phenotypic heterogeneity use simple on-off switches, recent results stemming from accurate, modern experimental methods require us to delve deeper into the dynamics. Here we provide a first step towards a more realistic theoretical framework for phenotypic switching.

When environmental fluctuations show stereotypical patterns, unicellular organisms may harness this temporal information to adjust their mode of phenotypic adaptation to match, or even anticipate, environmental fluctuations [1]. Such strategies can be embedded in genetic regulatory networks [2, 3] or arise from epigenetic phenotypic switches [4, 5]. These forms of fitness optimization by associative learning can arguably be assimilated to memory-based processes [6, 7]. Far from being a phenomenon involving cognition or learning, memory in bacteria may emerge as a component of phenotypic heterogeneity [8–12]. In homogeneous environments, non-genetic individuality can arise through fluctuations in the concentration of signaling molecules during transcriptional bursts [13], unequal partitioning at cell division [14] or via other epigenetic mechanisms [15–19]. When these events operate on molecules involved in ultrasensitive responses, they open the possibility that a population of genetically identical cells cultivated in homogeneous environment might split into two (or more) subpopulations of cells exhibiting qualitatively different phenotypes, often characterized by the on-off expression switch of some genes [20]. Numerous studies have identified bimodal distributions of gene expression levels in a variety of organisms, and it was shown that such 'in-built' mechanisms for phenotypic variation help bacterial populations adapt to harsh environments [19, 21–23].

Nevertheless, in most cases, the conditions and the dynamics at which cells alternate between different phenotypic states are still obscure. If switching from one phenotype to the other only relies on one stochastic event (that typically follows a Poisson process), one can predict that residence time in each phenotypic state should be exponentially distributed [24]. This hypothesis underpins a large body of theoretical work dealing with the evolution of phenotypic heterogeneity in bacteria [22, 25–27]. But this constant phenotypic switch, however attractive for its modeling simplicity, is not a generic model when approaching the problem of intergenerational memory. For example, it is inadequate when explaining phenomena such as broad time-lag distributions and both over- and undershooting behaviour of specific phenotypes observed in bacteria or eukaryotic organisms [4, 28, 29]. Recent technological developments now allow measuring actual switching rates and phenotypic residence times of individual cells grown in a stable environment. By studying the motility/chaining phenotypic switch in *Bacillus subtilis*, Norman et al. [30] showed that residence time in the motile state was exponentially distributed, whereas it followed a gamma distribution in the chaining state. Similar observations of non-exponentially distributed residence times are observed for cells exiting lag phases [4, 29]. Explaining these behaviours at the molecular level requires considering the entire

molecular network controlling the switch and its cellular context. For example, excitable gene regulatory networks can display activity pulses of stereotyped duration [31]. In the case of the *B. subtilis* motility/chaining switch, the high-affinity binding of two proteins forming an inactive complex, coupled to their dilution at each cell division, explain the timed pulses of chaining-associated gene expression [32]. In this context, memory can thus be defined as the ability of unicellular organisms to track the time already spent in a given phenotypic state. This sensitivity allows cells to adjust their switching rates according to some 'internal clock' so that the time spent in this state follows a gamma distribution [24]. In contrast, cases where the probability of switching remains constant with time, are deemed memory-less.

Working within this conceptual framework [24, 30], we propose a mechanistic approach to model memory that does not abdicate the simplicity of earlier work. It would be possible to directly implement memory by preselecting an appropriate non-exponential waiting time distribution. However, that does not help decipher a mechanism to how the distribution materialises. Hence, first, we derive deterministic dynamics from first-principles at the level of individual cells and track the behaviour of a cell population emerging from different switching genotypes. Using this model, we can consistently explain the experimental observation of over/undershoots and point to a possible way to understand wide time (non-exponential) lag distributions. Then, paying particular attention to the characterisation of transient states of the system, we discuss their potential consequences on the fitness of a lineage in the presence of fluctuating environments. Significantly, fluctuations across stressed and relaxed environments, as in specific antibiotic treatment regimes, highlight the applicability of our approach.

## Results

### Building memories

We simulate the ecological dynamics of isogenic populations of unicellular organisms that can exist in two different phenotypic states: 'on' and 'off'. Switching from 'off' to 'on' is unidirectional and stochastic and occurs at rate $\mu$. After turning 'on', cells cascade via a deterministic, multi-step process through $n$ compartments, eventually returning to the 'off' phenotype. These compartments represent internal molecular substates (potential), that may correspond to various biological processes, such as the dilution of cytoplasmic or membrane proteins [9, 11, 14, 32], or the sequential realization of independent molecular reactions collectively required to trigger a response [33–35].

Immediately after turning 'on', cells have the highest potential. While retaining the 'on' state, cells gradually lose potential by transitioning through the successive compartments at a leaching rate of $\epsilon$. In our model, this movement reflects a decrease in protein concentration in a simplistic manner, whereas more complicated forms can be formulated (with bumps or plateaus on the landscape, see S1 Text). Cell flow through compartments ($i = n, \ldots, 1$) is a representation of phenotypic dynamics, while growth (cell birth and death) dynamics occur separately at rates $b_i$ and $d_i$, respectively. Admittedly, birth and death can themselves lead to cell movement from one compartment to another (via symmetric or asymmetric division of the protein concentration in the daughters) [36]. We have explored such a model where cells differentiate into the downstream compartments separately and is available online on GitHub. However, to sort out the effects of leaching from growth and keep the model simple, in this manuscript, we have not included such scenarios.

Fig 1 visualizes this compartmental model for $n = 4$. The reactions in which an individual cell $X_i$ in compartment $i$ is involved are then,

$$
\begin{aligned}
X_i &\xrightarrow{b_i} X_i + X_i \\
X_i &\xrightarrow{d_i} \varnothing \\
X_i &\xrightarrow{\epsilon} X_{i-1}, \quad i = n, \ldots, 1 \\
X_0 &\xrightarrow{\mu} X_n,
\end{aligned}
\tag{1}
$$

These reactions have a deterministic counterpart in the following linear system of differential equations,

$$
\begin{aligned}
\dot{x}_0 &= (b_0 - d_0)x_0 - \mu x_0 + \epsilon x_1 \\
\dot{x}_n &= (b_n - d_n)x_n - \epsilon x_n + \mu x_0 \\
\dot{x}_i &= (b_i - d_i)x_i - \epsilon x_i + \epsilon x_{i+1},
\end{aligned}
\tag{2}
$$

which describe the time evolution of the abundances $x_i$ of cells in the compartments with $i = 1$, . . ., $n - 1$. We envisage that once a cell is in an 'off' state, it stays 'off' until an external or internal perturbation of sufficiently large magnitude switches it back 'on'. The origin of perturbations may be abiotic or biotic [21, 37]. For $n = 1$, we recover the classically studied memory-less switch. In what follows, we consistently compare this more classic model with two states and two compartments, which shows no memory according to our definition, with the case of our interest $n > 1$, i.e. two states yet multiple 'on' compartments, which can be shown to generate a form of memory that correlates with the magnitude of $n$. In the subsequent sections, we compare the performances of the two models ($n = 1$ versus $n > 1$) with particular attention to the transient phase. We ensure that the models can be considered equivalent from the point of view of long term properties by tuning the $\epsilon$ parameter accordingly.

However, we end this section by showing how our model meets the minimal requirement of producing memory. Tracking the amount of time a cell spends in a particular compartment makes this evident. Following the trajectories of individual cells, qualitatively different distributions of the time spent in the 'on' state emerge as the number of compartments $n$ increases. In particular, departure from an exponential distribution, the hallmark of memory [30] (Fig 1B) in this context, is observed. For a constant leaching rate and negligible growth parameters, the number of compartments acts like a timer (a deterministic time as termed in [30] for the residence time in the motile state). The overall density function can be captured by a combination of multiple exponential waiting times which results in a gamma distribution with a shape parameter given by the memory size $n$ (Fig 1B). This shows that our model satisfies the minimal requirement of generating memory as a deviation from an exponentially distributed waiting time before switching to 'on'. As the magnitude of this deviation relates directly to $n$, we conveniently refer to this parameter as the length or size of memory. Codes for implementing our algorithm as well as for reproducing the relevant figures are available at GitHub.

## Asymptotic properties

The asymptotic properties of the model were already well described and do not represent the primary focus of our study [30]. However, to make further progress, we recapitulate them briefly. As the more classic two-state two-compartment model, our model is linear and Markovian: the current system state entirely determines cell dynamics. The equilibrium cell distribution in the compartments can be recovered from the dominant eigenvector of the constant matrix that captures the system in Eq 2 (S1 Text). The corresponding eigenvalue corresponds

## (a) Dynamics

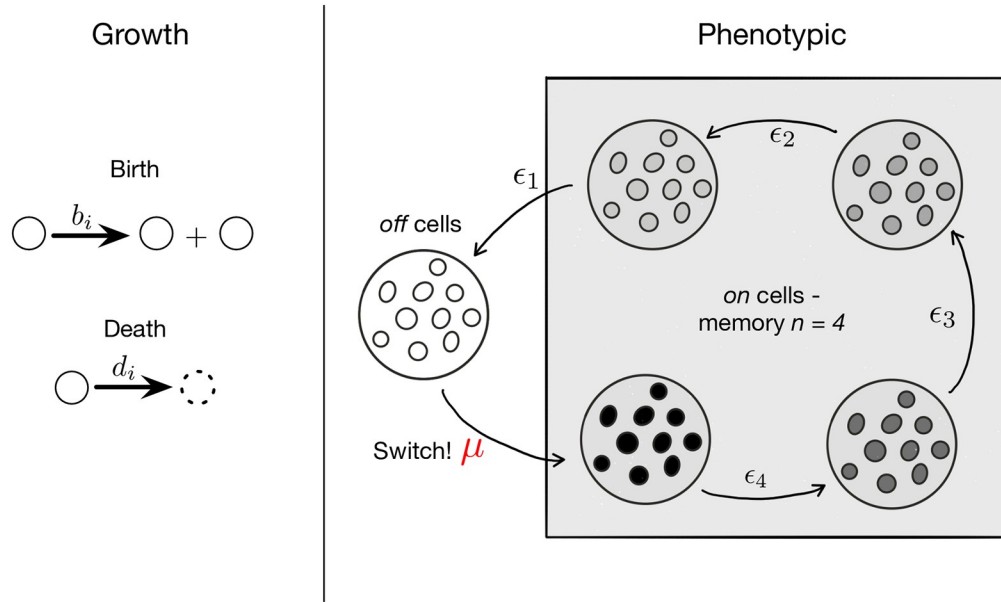

## (b) Waiting time distribution

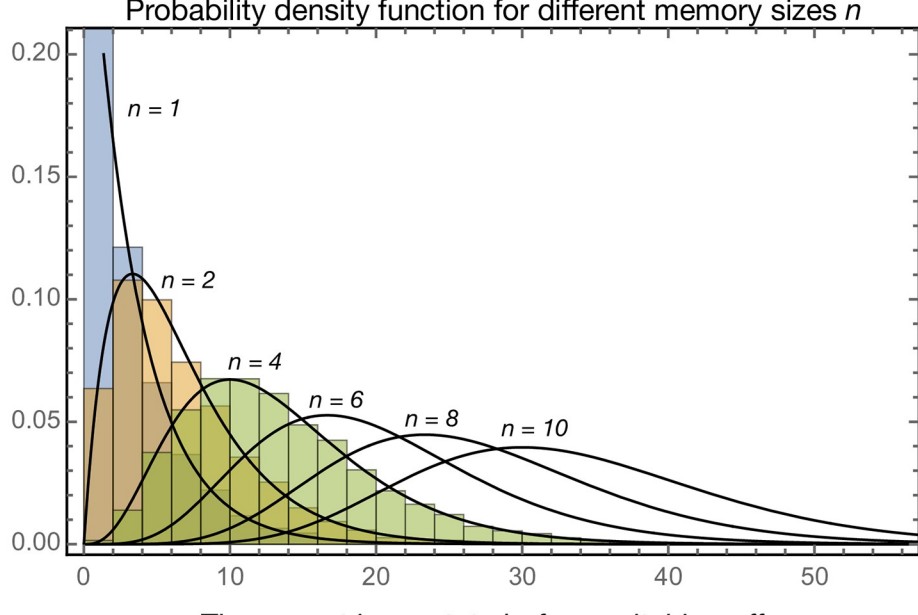

**Fig 1. Growth and phenotypic dynamics of 'off' and 'on' cells.** (A) The growth dynamics of cells is given by a birth process and a death process. A cell divides into two cells at rate $b_i$ and dies at rate $d_i$. The index $i$ refers to the phenotypic state of being 'off' or 'on'. The 'off' cells are the cells in the resting state which is the zeroth compartment $i = 0$. Due to a trigger (an internal constant or a dependence on frequency/density/environmental state) the cells can be turned 'on' at rate $\mu$ and they jump to the $n^{\text{th}}$ compartment (here $n = 4$). The cells do not stay in the 'on' state but slowly decay back to the 'off' /resting state via passaging at rate $\epsilon$ through a number of intermediate phenotypic stages, the different substates of the 'on' state. The number $n$ of compartments in the 'on' state modulates memory. With $n = 1$, there is zero memory, while for $n > 1$ memory increases in the number of compartments, i.e. the increase is here understood as the increase in

the average time spent in the 'on' state relative to the average of the exponentially distributed time spent in this state when $n = 1$. Thus we intend to make use of this mechanistic interpretation of memory instead of an assumption of a separation between the amount of time a cell spends in 'on' or 'off' state. (B) Distribution of times a cell stays 'on' before switching 'off'. Starting with a single cell in the completely 'on' state we ask how long it takes for the cell to reach the 'off' state. Assuming extremely low replication and death rates $b_i = d_i = 10^{-6}$, the process is governed by the leaching rates $\epsilon = 0.3$. For 1, 2 and 4 'on' compartments we start the Gillespie simulation with one cell in the first 'on' state. Once the cells reaches the 'off' state we stop the simulation. The normalised histogram of waiting times is then plotted. The probability density function of a gamma distribution with the shape parameter given by the number of 'on' compartments and the rate $\epsilon = 0.3$ provides the theoretical estimates (black lines) for the simulated 1, 2 and 4 'on' compartments. Increasing the memory, ($n = 6, 8$ and $10$) flattens the distribution of the time spent in the 'on' state.

to asymptotic population growth. Under complete symmetry in the growth dynamics ($b_i = b$ and $d_i = d$) we can get a simple, closed-form expression for this equilibrium for any number $n$ of compartments. We begin with a focus on this symmetric case, where the asymptotic growth rate $b - d$ of the population is independent of the number of compartments. The assumption is relaxed further in the manuscript. Our interest is in the effect of changes in the leaching rate $\epsilon$ and memory $n$. We observe that as $\epsilon$ decreases or $n$ increases, both the time a cell spends in the 'on' state and the equilibrium fraction of 'on' cells increases (see S1 Text). For a small number of compartments or faster transition through 'on' states, the system is dominated by 'off' cells.

## Transient properties

Bacteria in natural environments are frequently exposed to changing conditions. These fluctuations can often keep populations from reaching evolutionary or ecological steady states, and transient dynamics become crucial. Although less amenable to analytical treatment, we shall focus on them requiring greater reliance on computational exploration. While under constant switching ($\mu$), a decrease in $\epsilon$ is qualitatively equivalent to an increase in $n$ in terms of asymptotic properties (i.e. the equilibrium cell distribution and the time spent in either state), this equivalence breaks down when considering transient dynamics. As the number of compartments increases, the frequencies of the 'on' and 'off' cells do not approach their equilibrium values directly. Instead, in the case of 'on' cells, the frequencies overshoot and undershoot, in the case of 'off' cells. The frequency dynamics oscillates when approaching equilibrium (Fig 2A).

Interestingly, overshooting and oscillations in the transients of cell frequencies as seen in our model recapitulate observations in cancer cells dynamics [28] and in bacterial growth rate recovery following antibiotic treatments [38]. This transient effect directly relates to the length of memory. While modelling the 'off' and 'on' states as two compartments would appear more parsimonious, it falls short of replicating empirical results. Adding compartments intensifies transients, an effect attributable to the spectrum of the matrix model underlying Eq 2 (see S1 Text). The presence of multiple compartments introduces oscillations due to complex subdominant eigenvalues, absent when there is a single 'on' compartment. Increasing the number of compartments also magnifies the influence of complex subdominant eigenvalues on cell dynamics, as the real parts of such eigenvalues get closer to the real part to the dominant eigenvalue. Importantly, this analysis is possible while preserving linear dynamics and, thus, a form of simplicity. This, we suggest, would be lost if non-exponentially distributed waiting times were coded directly into a two-state two-compartment model via the introduction of non-linear terms in a coupled system of two differential equations.

Cells switch back to the 'off' state after spending a certain amount of time in the 'on' state. The presence of multiple compartments extends memory and affects population composition

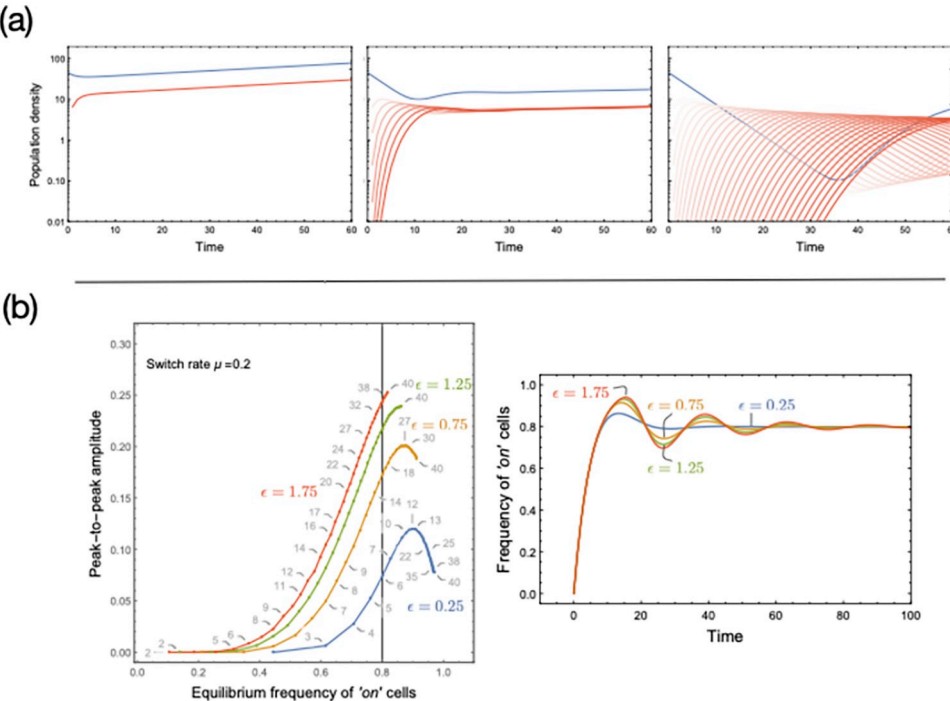

**Fig 2. Transient dynamics of multi-state memory.** *(A). Multi-state memory and overshoots.* Assuming full symmetry in growth dynamics between the compartments ($b_i = b = 1.0$ and $d_i = d = 0.98$) and a switch rate of $\mu = 0.2$ and $\epsilon = 0.5$ we show the population dynamics as well as the population composition for fixed time period. The instantaneous growth rate of the two population with different memory sizes is the same $g = 1/t_{max} \log(N_{final}/N_{initial})$. However, with a larger memory, the fraction of 'on' cells in the final population is higher. On the way to the equilibrium, multiple compartments result in overshoot dynamics as the 'on' cells first need to seed them while depleting the 'off' state. *(B). Transient magnitude and number of compartments.* To characterise the oscillations, we plot the peak-to-peak amplitude (the amplitude between the first peak, i.e. overshoot, and the second peak, i.e. undershoot) against the equilibrium value of 'on' cells as the number $n + 1$ of compartments increases ($n$ 'on' compartments and one 'off' compartment, gray numbers). The equilibrium fraction of cells in the 'on' state is given by $n\mu/(\epsilon + n\mu)$ (S1 Text). Keeping $\mu$ fixed, the peak-to-peak amplitude is obtained by varying the leaching rate $\epsilon$ and by varying the number $n$ of compartments in the multi-compartment system, which has constant $\epsilon$. As the equilibrium fraction of 'on' cells approaches unity, boundary effects prevail, i.e. transient frequency cannot exceed 1, and oscillations' amplitude gets compressed. The right panel shows the temporal dynamics of 'on' cells, all leading to the same equilibrium of 0.8 but for different leaching rate $\epsilon$ (vertical line in the left panel). The corresponding memory sizes are $n = 5, 15, 25, 35$ respectively. Increasing the number of compartments leads to more pronounced oscillations, not observed in the system with a single 'on' compartment regardless of the value of $\epsilon$ and the equilibrium value of 'on' cells. Codes for generating these panels and the general algorithm are available on GitHub.

as well as population density when evaluated at a specific time-point (Fig 2A). To capture the transient dynamics when increasing memory, we estimate the peak-to-peak amplitude (the difference between the largest overshoot and undershoot in the frequency of 'on' cells) (Fig 2B). Typically we see a monotonic increase in the relationship between the amplitude and memory length. The relationship wanes as the equilibrium value of 'on' cells approaches 1 due to boundary effects Fig 2B). More generally, the magnitude of the over/undershoot depends on the initial conditions and equilibrium of the system.

## Environmental variation

While cell lineages can stochastically switch phenotypes even in static conditions [39], such behaviour might be evolutionarily advantageous under fluctuating environments [40]. As the

environment changes, bacteria can hedge their bets via the well-documented phenomena of persistence [41, 42], a salient case being bacterial response to antibiotic treatment strategies.

To explore the potential effects of memory on bacterial fitness, we consider the case of persisters when subjected to transient antibiotic exposure. Persistence is a common phenomenon in bacteria where a subpopulation of bacterial cells does not grow but can survive antibiotic treatments. Persisters can arise stochastically via epigenetic switching, but environmental conditions (nutrient limitation, high cell densities or antibiotic treatments) can also induce their formation ('triggered' persistence or type I persisters; [37]). We track population composition and fitness (as proxied by population size) of bacterial lineages under fluctuating environments, consisting of transient exposure to antibiotics. We assign 'off' cells to the 'normal' physiological state (i.e. growing in permissive conditions, dying under antibiotic treatment). On the other hand, 'on' cells are persisters (no growth in either environment and tolerant to antibiotic treatment; see description of parameters in Fig 3 and example dynamics in S1 Text). While the leaching rate is kept constant, switching from 'off' to 'on' is triggered only under stress. We consider an initial population of 'off' cells, representing a stable equilibrium state under conducive growth conditions.

We subject a growing population of cells to a set of 84 environmental sequences where bacteria are exposed to drug treatment sequences (each horizontal line in Fig 3A corresponds to a different environmental sequence). The total duration of the procedure is kept constant, and the duration of drug treatment varies, with exposure to the permissive environment before and after drug exposure (also see S1 Text). Using these environmental sequences, we compute the fitness of a set of lineages with different memory sizes for each condition. We observe a non-linear relationship between memory size and fitness (Fig 3B). While there is a general trend that more memory is beneficial when drug treatment increases, local maxima emerge at intermediate drug treatment lengths (Fig 4). This effect arises from the trade-off between

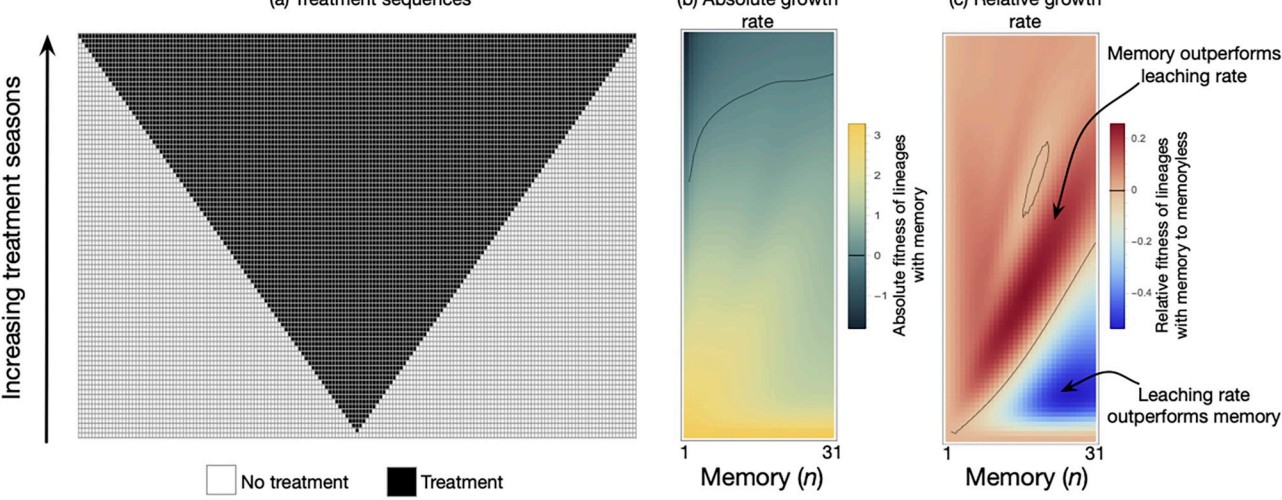

**Fig 3. Growth performance of lineages with different memory lengths exposed to treatment of varying length.** Cell lineages with different memory sizes ($n = (1, \ldots, 31) + 1$) are temporarily exposed to antibiotic treatment. Each row in the grid is a sequence of seasons from left to right. Seasons with no-treatment (white) and treatment (black) where each lasts for one unit of time. All sequences begin at $t = 0$ with 1000 'off' cells and last until $t_{max} = 165$. Under the no-treatment season the growth rate of 'off' cells is $b_{off} - d_{off} = 1 - 0.98 = 0.02$ and the switch is inactive $\mu_{no\ treatment} = 0$. As cells encounter the treatment season (black squares), the switch is triggered, $\mu_{no\ treatment} = 0.2$ and the death rate of 'off' cells increases, $b_{off} - d_{off} = 1 - 1.02 = -0.02$. The 'on' cells are produced but they do not grow, $b_{on} - d_{on} = 0$. The 'on' cells leach through the memory compartments at rate $\epsilon = 0.25$. At the end of the sequence ($t = 165$), the absolute growth rate, $g_m = \log(N_{memory}(t_{max})/N_{memory}(0))$, was computed. A memory-less process (two compartments with $n = 1$ and an appropriate $\epsilon$) can generate the same stable fraction of 'on' cells under sustained treatment. We subtract the absolute growth rate of such a memory-less lineage $g_{mless} = \log(N_{mless}(t_{max})/N_{mless}(0))$ from $g_m$ to estimate the relative fitness of having memory $r = (g_m - g_{mless})/t_{max}$.

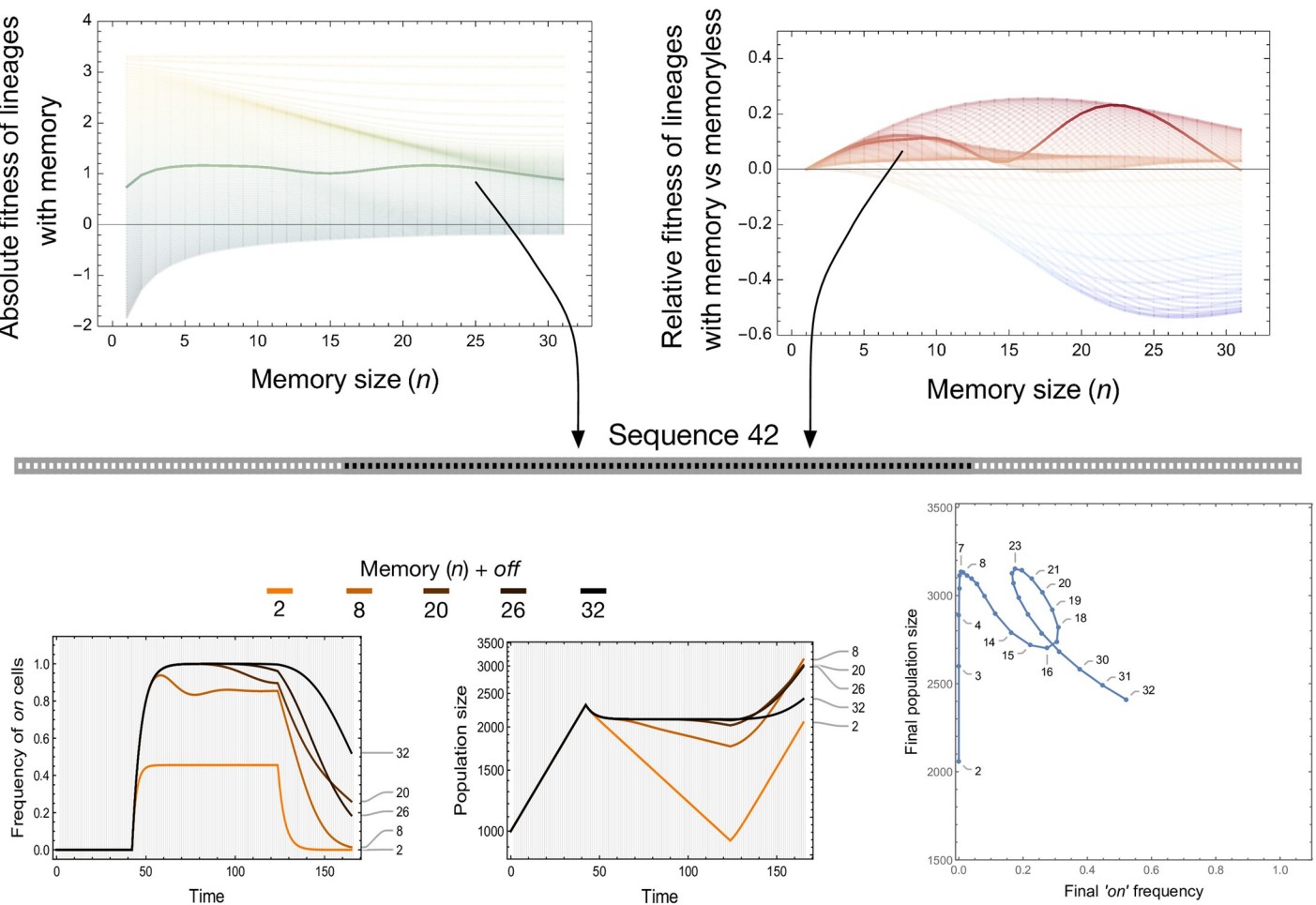

**Fig 4. Lineage adaptation under treatment.** Top panels show the effect of memory size on absolute and relative lineage fitness. Each of the lines in the top panels corresponds to each of the 84 sequences from Fig 3. The colorscheme of the left and right top panels is the same as from Fig 3B and 3C with the 42nd sequence highlighted in bold. We report the frequency of 'on' cells and the population size in the bottom panels. The results are shown for a chosen number of memory sizes (n) + 1 ('off' compartment). The non-linear relationship between the final 'on' cell frequency and the final population size is shown in the last panel as $n + 1$ ranges from 2 to 32.

producing a high proportion of 'on' cells (occurring with higher memories) and exiting rapidly from the 'on' state when the environment switches from drug to permissive (occurring with smaller memories). For extended treatment lengths, longer memory in such cases staves off an inevitable population collapse.

Memory is coupled to the transients and the final frequency of 'on' cells observed at the end of each sequence. To disentangle the respective effect of these two factors, we compared the fitness of a lineage with memory ($n > 1$) to a lineage without memory ($n = 1$). To make lineages comparable, the memory lineage has a different leaching rate which results in the same equilibrium 'on' frequency under sustained treatment Fig 3C. The relative growth rate is the difference between the growth rate of a lineage with memory and without memory following http://myxo.css.msu.edu/ecoli/srvsrf.html. When the treatment length is short, longer memory is disadvantageous as compared to larger leaching rates. The disadvantage arises because the cells stay in the 'on' state even after the short treatment has elapsed. For intermediate treatment

lengths, more memory allows (i) to produce more 'on' cells and (ii) longer residual time in the 'on' state, then becoming advantageous. However, long residual times corresponding to longer lag times could conflict with the total sequence length where the fitness is measured. Hence, very long memory is also not helpful as the cells then take much longer to exit the 'on' state than the sequence length. Thus, we see the presence of local maxima in memories (driven partly by the absolute fitness of the memoryless lineage, Fig 4). Under lengthy sustained treatment, all lineages are at the 'on' equilibrium, and memory length is inconsequential. This analysis reveals that memory outperforms 'classical' memoryless switching against an equal equilibrium value of 'on' cells for a range of conditions.

## Discussion and conclusion

Studies on phenotypic heterogeneity and persistence in bacteria have typically relied on models assuming two distinct cellular states, 'off' and 'on' with constant switching rates. We have argued that this assumption hinders us from understanding the rich dynamics observed in empirical studies. Keeping the two states model while decomposing the 'on' state into multiple, interconnected 'on' substates performs better in reconciling observations. The key to this reconciliation is that having more than one 'on' state generates memory. In this context, memory is the ability to revert to 'off' while accounting for the time spent in 'on' [24].

Multiple 'on' states are also implemented in a recent theoretical study studying the optimal switch time dependence on the likelihood and duration of a hazardous environment [43]. This study's overall approach was to look at evolutionary stable states but ignores transient population dynamics and stochasticity, an integral component of experimental findings. Our contribution fills this gap. We show that memory, originating from a process going through multiple substates, appears to explain otherwise anomalous observations.

Take, for example, the time it takes to resume division for a single cell taken out of a tolerant population that has just exited a prolonged antibiotic treatment [37]. This duration is the so-called lag time. Under artificial selection experiments, cells adapt their lag time to match treatment duration [4, 29] while displaying within-population variability. This variability increases with the mean lag time (see S1 Text). Initially, this observation seems counter-intuitive, as the best strategy would be for all cells to resume growth as soon as the treatment is over. Moreno-Gámez et al. [29] showed that heterogeneous lag times could promote survival to transient antibiotic treatments while having a negligible cost on population growth. In these studies, the lag time was shown to have a gamma distribution, which is not compatible with a memory-less process of switching from 'on' to 'off'. Our model finds that a cell in any 'on' state will exhibit a gamma-distributed lag time, coherent with observations.

It is possible to study how the lag time evolves in response to treatment likelihood and length [43]. Our framework, although not tackling this evolutionary problem directly, is suggestive of the fact that the variability in lag times upon which selection can act may potentially arise through multiple sources: changes in the number of 'on' states/compartments, in the leaching rate, in both or competition between lineages with variable properties of the gamma distribution. However, for a fuller understanding of the evolutionary forces acting on these traits, explicit considerations of their potential trade-offs will be required.

In experiments, the details of the growth dynamics often play an essential role. For example, cells may be first let grow exponentially, then starved into a stationary phase and finally sampled to regrow on abundant, fresh media. In our model, growth dynamics were minimized on purpose to focus on phenotypic dynamics. Moreover, while sampling from a single 'on' compartment would produce gamma-distributed lag times, sampling from multiple 'on' compartments would result in a lag time that is not gamma distributed, although emerging from a

combination of different gamma distributions (see S1 Text). Since our model abstracts bacterial populations' natural growth dynamics, we do not produce a qualitative match between the experimental data on lag times. To approximate experiments, it may be possible to design a two-compartment model with different waiting time distributions that recapitulate some of the properties that we present in this study. However, this would beg to reason the mechanisms resulting in the different distributions. Instead, we aim to postulate a mechanistic process that encompasses the internal "to-do" list before switching phenotype as suggested in [30]. Also, the compartments allow us to mathematically connect the number of 'on' states to the observed transient dynamics quantitatively.

Expressly, the transients of the dynamics can be understood in terms of eigenvalue analysis. Complex eigenvalues introduce oscillations in the dynamics (S1 Text). Larger memory sizes correspond to more solutions in the imaginary space, which are reflected in the dynamics with more oscillations (Fig 2). The magnitude of the effect hinges on how fast cells experience the memory (leaching rate) and the initial switch rate. For any leaching rate, however, as the memory increases, the oscillations (captured by the peak-to-peak amplitude) increase but only up to a limit (Fig 2B). The difference is solely in the fraction of the 'on' cells and not the population size. The result rests on our assumption of 'on' and 'off' cells having the same birth and death rates. Forgoing this assumption would lead to further divergence between the composition and size of lineages with different memories. Whether fitness depends on the composition or size of a lineage, memory will bring unique dynamical properties that might impact survival. The ecological context will set the timing of fitness evaluation, realising Darwinian selection on lineages (Fig 2).

Phenotypic heterogeneity while advantageous for a lineage [27, 44] can be a nuisance when population expansion is harmful. Since Hobby and Bigger [41, 42] persisters have been a fly in the ointment for antibiotic treatment only exacerbated now by the antibiotic crisis. We have presented the structure of the use of multi-state memory and its application to antibiotic tolerance. The persisters, as defined in our case, are a subpopulation of tolerant cells appropriately defined as "heterotolerant" [37]. While tolerance does not affect minimum inhibitory concentration, the duration of treatment will be crucial for eradicating bacteria. Another application of our approach could be in understanding the persistence and dormancy mechanisms in cancer populations when underlined by phenotypic switching [45]. Quiescent subclones are a persistent problem leading to cancer relapse [46–48]. The subclone population dynamics also show overshooting [28, 49]. A tunable and evolvable memory could then underlie the mechanisms of dormancy categorised as adaptive response [50]. Thus considering memory brings a different time-scale that can be exploited for steering evolution to control pathogenic populations.

Numerous extensions of our approach are possible. For example, we have focused on heterogeneity resulting from only environmentally triggered switches, a property observed in several experimental models [10, 12, 37, 51]. Alternative switching mechanisms may depend on the intrinsic properties of the population, such as density or composition. Similarly, the leaching mechanism is predetermined and constant across the compartments. Such processes may be under the complex, joint control of the organism as well as the environment. Although detailed experimental characterisation of bidirectional switching behaviours remains rare (due to technical challenges), we expect that memory-based switching is not an exception.

Oscillations involving excitation and decay can be identified across life. For example, excitable genetic switches are found in bacteria [52], belonging to circadian clocks in cyanobacteria [53], in other non-photosynthetic bacteria [54], in Drosophila [55] and in plants [56, 57]. The molecular mechanisms identified in these systems point towards a malleable duration of the oscillations by mutations [52, 58], modifications of the regulatory network [32] or degradation

and destabilisation of protein complexes [55]. Similar to the external cycling antibiotic stimulus explored in this study, the explicit dependence on external signals, flies' chronobiology has been proposed as a tool to understand adaptation to variable environmental conditions. Thus the adaptive evolution of molecular processes underpinning the oscillations of the excitable systems such as our model is conceivable.

While theoretical developments are essential, the applied aspect can be further exploited. Beyond antibiotics and cancer treatment, bioengineering and understanding microbial consortia formation could be informed using our approach. Knowing the memory limitations (e.g. coming from decay rates of protein complexes) involved in critical oscillatory processes such as in chronobiology can inform us about the limits of adaptation under extreme environmental events or anthropogenic changes. Especially when harsh environments and time-lags are of importance, such as in niche construction and the evolution of multicellularity [30, 59].

Understanding gene regulatory networks on a developmental landscape, à la Waddington [45, 60], poses exponentially complex computational challenges (e.g. the explosion of possible attractors when considering multiple switches [61] and multiple phenotypes) (as also discussed in [62]). We show multiple local maxima for memory, which may change depending on the definition of fitness (population size or composition). Furthermore, this forces us to rethink our concepts about possible phenotypic states and how they are determined by a plethora of molecular constructs (see S1 Text for an extended discussion). How epigenetic memory functions over generations would focus on understanding how the phenotypic clock (molecules, appropriate histone modifications) are inherited. In our model, time, as kept by degrading molecules, is arbitrary. Inputs from chronobiology could help see how the molecular concentrations and circuits are entrained, resulting in biologically driven growth and leaching dynamics.

Phenotypic heterogeneity forces us to reassess the genotype-phenotype map fundamentally. Choosing the appropriate phenotypic response to complex and varied environments is possible via numerous processes such as environmental sensing, epigenetic triggers and controlled molecular concentrations. Such processes that interpret genetics to a large but finite phenotypic space to survive in a possibly infinite environmental space are incredibly relevant for natural selection. Theories, as described herein, coupled with experiments exploring diverse environments, will help us elucidate the variety of possible interpreting mechanisms bridging the genotype-phenotype divide.

## Supporting information

**S1 Text. Supplementary text for memory shapes microbial populations.**
(PDF)

## Acknowledgments

We thank Silvia De Monte and Orso Romano for helpful discussions.

## Author Contributions

**Conceptualization:** Chaitanya S. Gokhale, Philippe Remigi.

**Formal analysis:** Chaitanya S. Gokhale, Stefano Giaimo, Philippe Remigi.

**Funding acquisition:** Chaitanya S. Gokhale, Philippe Remigi.

**Investigation:** Chaitanya S. Gokhale, Stefano Giaimo, Philippe Remigi.

**Methodology:** Chaitanya S. Gokhale, Stefano Giaimo, Philippe Remigi.

**Project administration:** Chaitanya S. Gokhale.

**Resources:** Chaitanya S. Gokhale, Philippe Remigi.

**Software:** Chaitanya S. Gokhale, Stefano Giaimo.

**Validation:** Chaitanya S. Gokhale, Stefano Giaimo.

**Visualization:** Chaitanya S. Gokhale, Stefano Giaimo, Philippe Remigi.

**Writing – original draft:** Chaitanya S. Gokhale, Stefano Giaimo, Philippe Remigi.

**Writing – review & editing:** Chaitanya S. Gokhale, Stefano Giaimo, Philippe Remigi.

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
