## [Decision Letter · Decision Letter 0]

1 Jun 2021

Dear Dr. Gokhale,

Thank you very much for submitting your manuscript "Memory shapes microbial populations" for consideration at PLOS Computational Biology.

As with all papers reviewed by the journal, your manuscript was reviewed by members of the editorial board and by several independent reviewers. In light of the reviews (below this email), we would like to invite the resubmission of a significantly-revised version that takes into account the reviewers' comments.

Thanks for your submission. Both reviewers saw valuable work in this manuscript. However, both also raise important concerns. I am inviting the authors to submit a major revision of the manuscript, taking into account each of the reviewer comments. Particularly important I think are the comparison and contrasts of this model with existing internal compartment models; the relevance or justification of terms like evolution and memory; and the explanatory value of large vs small numbers of compartments.

We cannot make any decision about publication until we have seen the revised manuscript and your response to the reviewers' comments. Your revised manuscript is also likely to be sent to reviewers for further evaluation.

Sincerely,

James O'Dwyer

Associate Editor

PLOS Computational Biology

Natalia Komarova

Deputy Editor

PLOS Computational Biology

Thanks for your submission. Both reviewers saw valuable work in this manuscript. However, both also raise important concerns. I am inviting the authors to submit a major revision of the manuscript, taking into account each of the reviewer comments. Particularly important I think are the comparison and contrasts of this model with existing internal compartment models; the relevance or justification of terms like evolution and memory; and the explanatory value of large vs small numbers of compartments.

Reviewer's Responses to Questions

**Comments to the Authors:**

Reviewer #1: In the present manuscript the authors analyze a model of persister cells (bacteria) which implements a mechanism that differs from the standard on-off switching. In particular, they assume that rather than having the usual two states "on" and "off", the persistent

state consists in a sequence of multiple intermediate "sub-states" or compartments

(at some constant leaching rate). This means that the cell needs to go through a succession of stochastic steps before being able to turn back to the awaked state. Not surprisingly, this combination of individual processes leads to a non-exponential distribution of lag times, i.e. a Gamma distribution whose shape is controlled by the number of compartments (while for the case of just one compartment, the standard exponential distribution is recovered).

Then the authors move on to analyze a situation in which a community of such cells -- either with many compartments, i.e. with "memory", or with just one of them (standard on-off scenario)-- is exposed to antibiotics during some variable time interval.

First of all, it is found that effects such as overshoots emerge only in the presence of memory (stremming from complex eigenvalues in the eigenvalues controlling the convergence to the equilibrium state. Then, the authors go on to show that there is a non-linear relation between memory length and absolute fitness; as could have been naively anticipated longer memories have a larger fit under longer antibiotic-exposure times and, conversely, for short antibiotic durations the tradeoff between survival and growth favors shorter memories. In particular, in this last case a strategy relying on a single compartment (with adapted leaching rate) can outperform memory, but, more generally, for a range of conditions memory outperforms classical memoryless switchning. They also elucidate a number of other properties of the model under different types of environments.

Overall the paper is clear, convincing and well written (well,... english should be corrected,

e.g. quite few "s" of verbs in third person are missing). Thus, in principle, I would be tempted to

recommend it for publication in PCB.

However, a number of important issues should be improved and clarified.

1) The authors do not cite the recent paper, published in PCB: "When to wake up? The optimal waking-up strategies for starvation-induced persistence", by Himeoka and Mitarai, that introduces a multistep process where "the cells sequentially go through multiple (M) dormant states" almost identical to the one proposed in the present manuscript. This work should be acknowledged and discussed.

2) When a cell reproduces the offspring are assumed to be in the some state as the mother. Is this a reasonable biological assumption? For instance, if the leaching rate is related to the concentration of some protein, I would be tempted to believe that the numbers of such a protein are divided somehow among the two offspring, rather than being perfectly duplicated; i.e there should be a reduction in the protein numbers. Putting this question in more general terms: the cells are transmitting their state to their progeny; is this supposed to reflect some kind of epigenetic transmission of information? In any case, this choice should be better motivated/argued and put into context.

3) The authors refer to the "evolutionary" part of the model as well as to the "eco-evolutionary" details. But, in the absence of mutation or variation in the inheritance process I do not see any evolutionary process here: there are just phenotypic changes and "internal state" transitions. Maybe "collective adaptation" of the community to the imposed external conditions would be a less confounding name.

4) Similarly, the term "memory" sounds nice, but at the end of the day what the model implements is a Markovian process. Moreover, "memory" evoques a learning process where some number

of compartments is selected after some environmental history. But here there is no such a

learning process. In my opinion, it is not well justified to call this effects a "memory"; even if I might be wrong, I think that this issue should be discussed more carefully in the manuscript.

5) This is up to the authors, but I found Figure S2 particularly clarifying: why not moving it to the main text and discussing it in more detail there?

Reviewer #2: In Gokhale et al, the authors describe an interesting and simple mathematical model of population-level phenotypic heterogeneity. In their model, cells can be in one of two states: a default OFF state; and when (stochastically) triggered, the cells switch to an ON state. The key ingredient in the authors’ model is the asymmetry in switching from OFF-to-ON versus ON-to-OFF. Specifically, the latter transition occurs slowly, as each cell transitions through a fixed number, n, of intermediate (but still ON) states, each at a leaching rate, epsilon, before they turn OFF. The authors claim that by tweaking n and epsilon, the two main parameters of the model, they can explain various phenomena observed in natural populations, such as the presence of antibiotic persisters, oscillations in the fraction of ON cells, and wide lag time distributions.

I certainly appreciate the model and I also agree with the authors that the role of memory in affecting microbial populations remains relatively under-explored. I think it will ultimately be a valuable contribution to the field. However, I have several concerns about the authors’ claims and results that I think need to be addressed before this paper is published.

Major concerns

1. Importance of multiple compartments: The authors claim that their model extends previous work by introducing several intermediate states (compartments) that cells must pass through while transitioning from ON to OFF, in this way going beyond models with just two binary states. While I understand this, I am still uncertain to what extent the authors need a model with multiple compartments to explain the key takeaways of the paper. In my understanding, even with just two compartments, the authors could have directly controlled the waiting time distribution, both its mean and width, and obtained very similar results. Moreover, it would still take the same number of parameters: two (one each for a mean and width) to describe such a model. I kept wondering why a multi-compartment model was necessary while reading the paper. I could argue about most results intuitively (the presence of persisters, for instance), and even the most interesting result (the phase diagram in Fig. 3) was something that seemed to work with a two compartment model in my head.

The authors claim that “adding compartments intensifies transients”, but as I will discuss in a later point, this part of the paper needs some work, especially (for me) being better motivated biologically.

I therefore invite the authors to more carefully and explicitly explain which aspects of their results require multiple compartments, and which aspects can suffice with just two compartments, through direct control of the waiting time distribution in switching from the ON to OFF states.

2. Biological basis for microbial memory: The authors argue that phenotypic memory could be realized via distinct internal states, such as through the concentration of an internal molecule. This to me needed much more discussion and a lot of biological motivation throughout the text. While I find it easier to imagine memory being realized through multi-cell circuits in eukaryotes, I find it quite hard to think about them in unicellular populations. More direct examples, such as discussing specific studies and putting them in context would really help.

I therefore invite the authors to juxtapose the results of their model with real biological observations not just throughout the text, but also more explicitly, by discussing the relevant numbers and representing the observed phenomena in the figures. This could be done, for instance, by showing real lag time distributions and comparing their features with those produced by the model.

Further, the authors should give some concrete examples of realizing memory and relating it with microbial phenotypes. How should one think of multiple discrete phenotypic states in the case of antibiotic persisters? Which molecules should we think of as examples? I understand that the model is quite general, and as such does not describe a particular molecule or cell, but it would be nice to ground it in the context of a few real and biologically motivated examples.

3. Transient damped oscillations: The section titled “Transient properties”, to me, represents the biggest opportunity for the authors. Here, they can talk about the damped oscillations and show what they look like in real systems. In the Discussion, the authors talk about circadian clocks and other oscillations, but in my mind they are quite different from such damped oscillations. I think the mechanism of how the overshoots and undershoots occur should also be explained somewhat more through cartoons in Fig. 2, rather than just showing the phenomenon. Moreover, this figure generally needs work, because I could not find the relevant legends anywhere, so I didn’t know what the colors in Fig. 2A mean.

Additionally, in general, since the authors claim that multiple compartments are crucial for such transients to occur, I think this section needs to do a lot more to convince the reader that this really is a phenomenon we need to explain, and that two compartment models of various kinds are insufficient. For instance Fig. S1 shows that with n=1 compartment, one does not obtain transients, but that’s when there is a fixed leaching rate, epsilon. How about the two compartment model I described, where the mean and variance of the waiting time distribution can be tuned independently? Would that be able to obtain transients? I am sorry for belaboring this point, but I would really like to know, since I think the authors have something interesting here.

Minor concerns

1. What do the curves in Fig. S7 represent? Are they trajectories? If so, where do they start? I had a hard time understanding this figure, but I also think it might be potentially interesting. The reason is that it is still unclear to me whether, for memory to be beneficial, the antibiotic episode needs to be one long sustained episode, or whether it is the total amount of time the antibiotic was present that matters. Fig. 3 alone cannot answer this, but Fig. S7 might. Naively, I would think that the duration of an episode is important, since that is what you need to tune, but if you have multiple intermittent episodes very often, then it might be useful as well.

----------

Akshit Goyal

Massachusetts Institute of Technology

**Have the authors made all data and (if applicable) computational code underlying the findings in their manuscript fully available?**

Reviewer #1: Yes

Reviewer #2: Yes

PLOS authors have the option to publish the peer review history of their article (what does this mean?). If published, this will include your full peer review and any attached files.

Reviewer #1: No

Reviewer #2: **Yes: **Akshit Goyal
---

## [Decision Letter · Decision Letter 1]

8 Sep 2021

Dear Dr. Gokhale,

We are pleased to inform you that your manuscript 'Memory shapes microbial populations' has been provisionally accepted for publication in PLOS Computational Biology.

Best regards,

James O'Dwyer

Deputy Editor

PLOS Computational Biology

Natalia Komarova

Deputy Editor

PLOS Computational Biology

Reviewer's Responses to Questions

**Comments to the Authors:**

Reviewer #1: The authors have improved the paper significantly following all the suggestions.

It can now be accepted for publication.

Reviewer #2: The authors have adequately responded to and handled my concerns, for which I thank them. I believe the manuscript is now suitable for publication.

**Have the authors made all data and (if applicable) computational code underlying the findings in their manuscript fully available?**

Reviewer #1: Yes

Reviewer #2: Yes

PLOS authors have the option to publish the peer review history of their article (what does this mean?). If published, this will include your full peer review and any attached files.

Reviewer #1: No

Reviewer #2: **Yes: **Akshit Goyal

---

## [Editor Report · Acceptance letter]

28 Sep 2021

PCOMPBIOL-D-21-00456R1 

Memory shapes microbial populations

Dear Dr Gokhale,

I am pleased to inform you that your manuscript has been formally accepted for publication in PLOS Computational Biology. Your manuscript is now with our production department and you will be notified of the publication date in due course.

With kind regards,

Andrea Szabo
